# Forbidden, yet common: Female genital cutting among the Oromo in central Ethiopia

**Zerihun Mekuria Tesfaye, Dejene Gemechu Chala** 🅾 *, **Jira Mekonnen Choroke**

Department of Social Anthropology, College of Social Sciences and Humanities, Jimma University, Jimma, Ethiopia

* sachekebo@gmail.com

## Abstract

This article answers the question 'why female genital cutting (FGC) has remained common among the Oromo in central Ethiopia when it is legally forbidden.' Primary data for this article were generated through key informant interviews and focus group discussions, while secondary data were gathered from online and archives in the research site. Our findings indicate that the government tries to use legislative and health approaches in its efforts to terminate FGC. However, it has survived over time because of the strong resistance that the community in the research area wages against the intervention efforts. The very reason behind the resistance to end FGC is a patriarchal value that shapes gender relations, women's sexuality, and behaviors. The community gives precedence to the socio-cultural values of FGC over its criminalization and the stated negative health effects. Paradoxically, women are at the forefront in promoting FGC, for they believe that FGC shapes the sexual behavior of the girls and defines the gender identity of their daughters. The women take the prime responsibility to raise and mold the girls into ideal female behavior.

**Data Availability Statement:** Nearly all relevant field data is included in the manuscript. However, thirteen FGC-related cases were collected, and some of them are missing from this document. We have uploaded the remaining cases as a separate

## Introduction

This article focuses on female genital cutting (FGC) among the Oromo of Dawo District in Oromia National Regional State (ONRS), central Ethiopia. FGC includes all the surgical procedures for partial or total removal of external female genitalia and other injuries to the female genital organs for no medical reasons. It ranges from a minor incision at the tip of the clitoris to a total excision of the external genitalia and stitching of the vaginal opening [1, 2]. This article, thus, addresses the question of how and why FGC, which is legally forbidden and is said to be harmful in terms of health and a violation of human rights, has remained common in the study area. The community challenges the criminalization of FGC and other interventions by the government, mainly by maintaining the confidentiality of the practice and undermining the penalties that result from the practice.

A recent publication (e.g. [3]), which focuses on the Dawo District, shows that more than 90% of the schoolgirls who participated in the study at the time had undergone FGC. UNICEF [4] also shows that the prevalence of FGC in the ONRS, where our research site is situated, is between 76% and 90% for women between the ages of 15 and 49.

file under the file category 'supplementary data.' In addition, the present manuscript is derived from a larger research project titled "Gender and Human Rights of Women," as specified in the ethical clearance letter provided by the review board of the host institute. We have audio versions of our first-hand information about the wider project; these are clearly outside the scope of this specific manuscript. The audio data is not only large but it is not meant to be used simply for this paper. We also obtained data from the police department and district court concerning prosecutions for FGC offenses under Ethiopian law. These data were retrieved from where they were located and examined in the manuscript. We could only access these data at their locations due to the very nature of these offices. In addition, the semi-urban administrative center, Busa, where these two offices are located, lacks basic infrastructure like an internet connection and a P.O. Box of their own. As a result, there are no official websites or data repositories where they may keep their files. That means the data we gathered were available only in hardcopies, and the offices in focus have physical addresses, P.O. Box that belongs to the neighboring district, and telephone numbers. Thus, we have provided the P.O. Box and telephone numbers of the two offices. 1. Dawo District Police Office Southwest Shoa Zone Administration Oromia National Regional State Ethiopia P.O. Box 11 (belonging to Bacho District, Tulu Bolo Town) Tel. +251 113120041 2. Dawo District Court Office Southwest Shoa Zone Administration Oromia National Regional State Ethiopia P.O. Box 11 (belonging to Bacho District, Tulu Bolo Town) Tel. +251 113120054 Finally, this manuscript originated from a staff research project that was owned by the College of Social Sciences and Humanities of Jimma University. The university is also responsible for ensuring the ethics of the research. We, thus, have provided the address of the college for your reference. College of Social Sciences and Humanities Jimma University (official website: http://www.ju.edu.et) Ethiopia P. O. Box: 378 Tel: +251471122432

**Funding:** The author(s) received no specific funding for this work.

**Competing interests:** there is no competing interest

The World Health Organization (WHO) estimates that globally, more than 200 million girls and women are affected by FGC, while an additional three million girls are at risk of having their genitalia cut every year. The magnitude greatly varies across regions, while the high concentration of this practice is in some African, Asian, and Middle Eastern countries [5, 6].

In Ethiopia, as in other African countries and beyond, FGC has remained a common practice. Despite its significant variations in terms of prevalence and severity, the practice is generally widespread across all regions, religions, and ethnic groups. A study (e.g. [7]) shows that 24% of Ethiopian girls between 0 and 14 years of age have undergone some form of FGC. About 74% of the women between the ages of 15 and 49 have already had their genitals cut. A report by another study [8] rather reduces this percentage to 65. In terms of regional variation, according to some sources, about 24% of women aged 15–49 have already undergone FGC in Tigray National Regional State, whereas the percentage is 98.5 for the Somali region in Ethiopia. With reference to types, FGC ranges from clitoridectomy to infibulation and even beyond, including pricking, piercing, incising, scraping, and cauterization [9, 10].

Nonetheless, FGC is a criminal act in the country. The Ethiopian Criminal Code [11] states, "Whoever circumcises a woman of any age is punishable with simple imprisonment for not less than three months or a fine of not less than five hundred Birr." Article 566 (1) reads: "Whoever infibulates the genitalia of a woman is punishable with rigorous imprisonment from three to five years." In 2013, Ethiopia introduced a National Strategy and Action Plan to eliminate harmful traditional practices (HTPs) that include early marriage, child preference, wife inheritance, abduction, uvulectomy, and milk tooth extraction against women and children. FGC is also one of these traditional practices, which the country aims to eliminate [9, 12]. Despite the national legislation being in place, the practice has continued mainly because of some challenges related to awareness of the people, associated cultural values, and law enforcement measures. Local community leaders and many officials strongly support the continuation of FGC more than they value the law banning it [9].

However, the question of whether or not the legal enactment of the prohibition of FGC as well as the strong campaign against it are effective in ending the practice demands due consideration. Thus, this article aims to explain why FGC remains a common practice among the Oromo in Dawo District, Oromia National Regional State, Ethiopia. To answer this question, the article tends to show that the community gives precedence to the cultural values of the FGC over its health effects and criminalization. It also addresses ways to withstand the potential legal prosecution that results from the FGC. We argue that the patriarchal system is behind the scenes to promote FGC in this research area. Below, we briefly discuss some concepts related to this FGC to situate it in the wider literature and to show factors contributing to its continuation in this area.

## FGC: A conceptual definition

FGC as a cultural practice takes many forms, even within the same community and society. There are numerous terminologies for FGC. These are "female genital mutilation," "female genital cutting', "female circumcision," "female genital surgery," "female genital modification," and so on [1, 13]. The multiplicity of terminology indicates the complexity of the problems related to FGC. These different terminologies reflect the different positions or contexts in which the issue is being understood [1]. Debates around the terms and the meanings they convey are ongoing [13]. However, a related study [7] shows that the use of the term "female circumcision" received grave criticism for tending to normalize the act by comparing it with males' circumcision that is a widely accepted practice. Since the 1970s, feminist activists have exposed the harmful consequences of this tradition and began to use the phrase "female genital

mutilation" (FGM) instead of "female circumcision." On the other hand, some countries insist on using the term "female circumcision," since the term "mutilation" connotes offensive and evil intent. "Female genital cutting" is also another alternative term used in the literature.

At our research site, *dhaqnaqabaa*, (literally, 'catching the body'), which connotes 'circumcision,' is the compound word that refers to female genital cutting. The official Ethiopian legal tools also use the term 'circumcision', for FGC. Our target population practices what is commonly known as clitoridectomy, that is, the removal of part of the clitoris. In Ethiopia, there are four forms of FGC, namely: clitoridectomy, excision, infibulation, and all related harmful procedures like pricking, piercing, incising, scraping, and cauterization [9]. Infibulation is more severe and common among the Somalis and the Afar peoples, while in Oromia, clitoridectomy is more common [4, 9, 14]. In this article, we use the term "female genital cutting" (FGC) to be less judgmental about this culturally-sensitive term [15]. The following section focuses on some of the common values behind FGC and the approaches/efforts that are being used to terminate it.

## The cultural values of FGC

There are a variety of explanations behind the practice of female genital cutting. Some of these explanations have socio-cultural, hygienic, aesthetic, and psychosexual reasons. The practice serves as a rite of passage that turns girls into womanhood. By removing part of their genitals, the clitoris, which has the appearance of a male sex organ, FGC also serves as one way of ensuring a girl's gender identity that results from the modification of the sex organ [16–18].

Some studies (e.g. [13, 19]) explain that patriarchy provides the cultural and institutional framework for the perpetuation of FGC. It is a reflection of the power inequality between men and women, in which men control women's sexuality, their reproductive systems, and their health. In short, FGC persists because of patriarchy. Pertinent to this, as a study (e.g. [20]) suggests, the concept of "clitoral economy" shows the symmetry between the female body (the clitoris) and the male body (the penis). The author shows how the clitoris is vested with so much power in symbolic and semiotic constructions of the female body. Clitoris has been viewed as a threat to the hegemonic phallic economy in patriarchal societies where FGC is common.

In this regard, there is a paradox behind FGC in patriarchal societies. Women who are victims of the practice defend and promote it as part of their tradition, which marks a rite of passage as well as a cosmetic procedure with some aesthetic values [13]. In some African cultures, sexual purity presumes FGC and is taken as a precondition for marriage. In Somalia, where FGC is mandatory and any girl who has not been exposed to it will never marry, is the case in point. The second reason is to attain a feminine body by removing the erectile part and to shape it into a socially-accepted behavior as a result of the practice [21].

In traditional African culture, older women reproduce patriarchal traditions by imposing themselves on younger girls, the same way they were treated before. In this regard, women adjust themselves to be active agents within the patriarchal system to promote the system that has suppressed them [22–24]. However, some studies show how the role of women in sustaining FGC changes across socio-demographic factors. There are correlations between FGC and parental factors, particularly maternal education, age, FGC experience, and residence setting. The more mothers are young and educated and have undergone FGC, the less likely it is that their daughters will experience the same [25–27].

## Approaches to ending FGC

The UN, WHO, UNICEF, and other women's rights advocacy organizations have used four main approaches to end FGC. These are bodily and sexual integrity, human rights, legislative,

and health approaches [28]. Feminist writers who advocate a bodily and sexual integrity approach emphasize women's sexual rights and satisfaction. This approach underlines how FGC impairs women's right to sexual health and pleasure by shaping, mediating, and controlling women's sexual enjoyment through social institutions [29]. In this regard, functionally, the clitoris is a site for sexual pleasure and is taken as a symbol of female sexuality by Western "second-wave feminists," and the removal of this part of women's genitalia obliterates their sexual enjoyment, freedom, and expression [28]. In this regard, FGC is also linked to the need to reduce promiscuity and maintain premarital virginity [14, 28, 30].

However, some challenge the bodily and sexual integrity approach because they believe that sexual enjoyment must be "normal" and ethical. As a study suggests (e.g. [28]), since sexuality is guided by societal norms and is a private issue, conservatives challenge the Western prescription to stop FGC. In conformity with this view, a study (e.g. [31]) reported that about 30% of Kenyan women are in favor of the continuation of FGC to preserve virginity.

The second is the human rights approach, which considers FGC as a basic violation of the human rights of girls and women. According to this approach, FGC harms females' bodies for no medical reason, and it is a gendered discrimination and a fundamental violation of the human rights of women and children, including the right to freedom from discrimination, torture, and violence; the right to health, the right to education, and the right to life. This approach is prominent with the UN, the European Union, and several governments of developed countries [28]. FGC violates the rights of girls who are not able to consent to the practice and has to, therefore, end [6, 14]. This position is what is usually known as the "zero tolerance" approach of the agencies [1].

However, there are also challenges to the human rights approach, for it is biased toward the liberal democratic tradition that gives priority to individual rights instead of "community" rights [31]. For instance, Africans perceived the 1960s and 1970s European and American feminists' campaign to eliminate FGC as an imposition from the outside and another attempt at colonial imperialism [28].

The legislative approach is the third-to-end FGC. Since the turn of the century, governmental, intergovernmental, and non-governmental organizations, as well as academia, have made FGC a hot topic. Most of the countries where FGC is common have laws that prohibit the practice [13, 32, 33]. For instance, the majority of African countries have laws that criminalize it. The African Charter on the Rights and Welfare of the Child is another effort to abolish the practice [13]. These laws have labeled FGC as a criminal act of bodily harm or injury and provided a structural framework or "enabling environment" to act against it [28].

However, the debate is whether it is possible to end FGC by criminalizing the practice or not. There is a consensus that the law alone does not serve the purpose. Despite the criminalization of the act, local practitioners continue to cut female genitals secretly. This practice has contributed to even more unsanitary and unsafe conditions during the surgery [13, 17]. There is another negative side to the legislative approach for parents who practice FGC underground, and it is less likely that they seek health care for complications resulting from the surgery for fear of prosecution [28].

The health approach is the fourth one, which emphasizes the negative health implications of FGC [13]. The WHO focused on health hazards to justify strongly the abolition of FGC. However, later, UN agencies shifted their attention from bodily injuries and therapeutic risks of FGC to the human rights violations in their efforts to end it [1]. Written sources show that the practitioners include midwives, trained circumcisers, doctors and nurses, or traditional practitioners—local specialists (circumcisers), traditional birth attendants, traditional healers, and experienced grandmothers—without any formal training. Regardless of who the practitioner is, she or he does the surgery mostly without anesthesia, antibiotics, or sterile equipment

[7, 31]. Thus, the practice has short- and long-term health effects. The short-term impacts can include severe pain, shock, bleeding, tetanus or bacterial infection, urine retention, open sores in the genital region, and injury to nearby genital tissue. The long-term consequences include recurrent bladder and urinary tract infections, cysts, infertility, an increased risk of childbirth complications, and the need for later and repeated surgeries for childbirth and sexual intercourse [6].

However, this is not without controversy. Some challenge this approach, for it does not take into consideration the lived realities of women who have experienced different forms of FGC [28]. In some cases, the community has considered FGC an important surgery. For instance, the Nigerians and the Ghanaians believe that FGC cures infertility, and for some Sudanese, it prevents diseases in infancy and promotes child health [13].

Despite the fact that the above approaches are all relevant to put an end to FGC, the local government at our research site uses the legislative and health approaches more vividly. The Ethiopian criminal code [11] labels FGC as a criminal act and a violation of the human rights of women and girls. There are also various government offices working against FGC and other cultural practices that are categorized as harmful. These are the Women, Children, and Youth Affairs (WCYA), Culture and Tourism, Health Office, Education, and Police offices. Despite this, FGC survived as a longstanding cultural practice. In the following sections, right after methods and materials, we will look at the cultural meanings of FGC as well as the factors that contribute to its persistence in the face of the government's efforts to end it. Thus, the objective of the study was to discuss how FGC has survived, although there are legal instruments and institutional setups that work towards the end of the practice

## Methods and materials

### Study site

The study was conducted in Dawo District, which is situated in central Ethiopia. The administrative center of the district is located approximately at 90 kilometers southwest of Addis Ababa. The total population of the district is 122,731 people [34], while the Oromo constitute 93.35 percent of the total, and Afaan Oromo (Oromo Language) is both the official and spoken language of the district. Ethiopian Orthodox Christianity is practiced by more than 90% of the population. There are followers of traditional beliefs and Protestant Christianity [35].

### Research design

The study employed a qualitative approach in which both primary and secondary data were used. Data from the field were collected through key informant interviews and focus group discussions. The fieldwork was conducted from February 27 to March 20, 2020. All the participants of the research were purposively selected in consultation with the office of the WCYA of the district. Partly, convenience sampling was used to identify the participants. The selection criteria were individual knowledge and access to information about the issue under discussion. For the secondary data sources, the Ethiopian Federal Constitution, legal codes, and policy strategies were retrieved and used, and documents from the district's WCYA and police station and closed files from the district court were used to show some government efforts to stop FGC. These secondary sources, such as the constitution, legal codes, and policy strategies, were accessed online, while others were secured at their respective places in the research site. Generally, the secondary data were used to supplement the primary data, and the former shows the normative standards while the latter is more of community practice from where our title was drawn.

### Key informant interviews

We interviewed 20 key informants—13 women and seven men. The participants were three (a man and two women) from the Women, Youth, and Children Affairs Office; a focal person from the Police Office (a woman); an expert from the Culture and Tourism Office (a man); an expert from the Education Office (a woman); four women and three men from the community; and seven schoolteachers (five women and two men). Three of the five women were married and had children, while the rest were bachelors. The two men had daughters. The offices listed above were chosen because they have directly or indirectly worked on the abolition of the FGC. Elderly, male, and female key informants were purposively chosen from the community to gather data on the cultural values underlying FGC and the community's reactions to government interventions to abolish it. The key informant interviews took between thirty minutes with community members and ninety minutes with an expert from the office of WCYA.

### Focus group discussion

We also used focus group discussions (FGDs) to triangulate the data from key informant interviews. To look at the concerns of different individuals and to explore their common understanding of some issues, we organized two FGDs. In the first FGD, there were eight experts (four men and four women) from concerned offices: two from the office of WCYA, two from the Culture and Tourism office, one from the Education Office, a schoolteacher, and two from the district police. The participants of the second FGD were nine married women who had experienced FGC as recipients and mothers. This method helped to access data from individuals as members of a group rather than simply as individuals. The discussions, debates, and consensus were all important because, by bringing together different individuals who have a common background, FGD plays an essential role in building meaningful data from common societal issues and confrontations.

### Data collection tools and analysis

For both key informant interviews and FGDs, we used semi-structured questions that were few in number and open-ended in nature, with the objective of eliciting views and opinions from the participants. These questions were set with the intention of guiding the informants rather than restricting them to answering what was just asked. All the interviews and FGDs were conducted in Afaan Oromo, the language, which we, the researchers, speak natively, and the principal investigator is well familiar with the research site. We transcribed the data into the original language and translated it into English. We used a thematic qualitative data analysis whereby we coded, categorized our transcript, interviews, and secondary data into themes and sub-themes, and finally interpreted them qualitatively.

**Ethical consideration.**   Before conducting the study, ethical clearance was obtained from the Research and Ethical Review Board (RERB) of the College of Social Sciences and Humanities of the hosting university (the name of the university temporarily remained confidential for article reviewers). After reviewing the proposal and specifically the participants 'consent form to be signed, the RERB fully authorized the conduct of the research. The research did not involve minors, and it was conducted by implementing all human handling during the data collection. All the participants were informed about the purpose of the study and provided their consent in written form to participate in the research. The written consents were for the consumption of the researchers and RERB. Finally, the findings of the study were reported in an aggregate fashion, and codes (36–59) were used to quote participants. Personal names are not mentioned in this article.

## Result

### Approaches to end FGC and community resistance in Dawo District

The micro-level efforts to end FGC are direct reflections of the national policies, rules, and regulations. The major macro-level approaches are the human rights approach, the legal approach, and the health risk approach. As indicated in the introduction, Ethiopia introduced a penal code banning FGC (Ethiopian Criminal Code Article 565 and 566). In addition, in 2013, a National Strategy and Action Plan on Harmful Traditional Practices against Women and Children, which was followed up by another strategy known as the Social Norm Change Communication Strategy, were made effective. The strategy to end HTP has been based on the three-pillar approaches of prevention, provision, and protection [12]. Led by the Ministry of Women, Children, and Youth (MoWCY) the country set up a national alliance to terminate child marriage and FGC. Other sectorial ministries, national associations, faith-based institutions, international and local civil society organizations have collaborated on the implementation of the national strategy. In addition, the Ethiopian government has restated its plan to stop FGC and child marriage by 2025 [10].

In January 2017, the Ministry of Health banned the medicalization of FGC in all public and private medical facilities, and any breach of this ban will be subject to legal action. The restriction goes hand-in-hand with the criminalization of FGC under the Criminal Code of 2005 [9, 10]. As an extension of the criminalization of the act, the district police office has a focal person who works to protect the human rights of women and children. The focal person is in charge of handling cases such as early marriage and/or abduction, rape, and violence against women, including FGC.

The MoWCYA at the federal level has offices at regional, zonal, and district levels to implement national policies and strategies to end FGC and to ensure gender equality (the Ethiopian federal state has regional, zonal, district, and *kebele* levels of administration) [10]. There is a district office for WCYA, designed to oversee and protect the rights and well-being of women and children, and FGC is one of its central focuses. The Office of Culture and Tourism is the one in charge of promoting cultural elements and fighting against "harmful traditional practices." The schools are also responsible for teaching the young generation to develop all-rounded personalities. They attempt to create awareness about the adverse effects of practices such as FGC and early marriage. These offices and institutions work in cooperation and even as a team against the FGC [36, 37].

The national FGC network that aimed at fighting FGC was established in Ethiopia in 2002 under the initiative of Norwegian Church Aid. The network was officially launched in 2010 and had 46 members from government, NGOs, faith-based and civil society organizations, and UN agencies (notably UNFPA, UNICEF, and WHO) [14]. Ethiopia also introduced a health extension package to reach villages and households through health extension workers. Among others, the health extension workers focus on creating awareness to end harmful traditional practices (HTPs), including FGC.

### The persistence of FGC in Dawo District

Data from our informants [36–39] show that the community is well-informed about the criminalization, the negative health impacts, and human rights violations related to FGC. However, mere knowledge of the issue has not deterred the community from practicing FGC. The primary data from the field imply four explanations for the persistence of this practice. Primarily, the community gives precedence to the cultural values of FGC over its negative health effects and its legal implications. Secondly, the community has designed a mechanism to escape the

potential legal prosecution that could result from the violation of the law that bans it. Thirdly, the community generally undermines the health effects of FGC by appealing to their lived experiences. Fourthly, the community also disregards the fines that could result from the practice. On the other hand, issues related to individual rights, bodily integrity, and sexual pleasure are not a point of attention for the community. That means individual rights has no space in this community because it is collectivistic, focusing on 'we' consciousness, collective identity, and emotional dependence rather than the opposite. The norm does not also allow the discussion of the second issue,—sexual integrity,—in public. In the following sections, we focus on the explanation of these issues.

## Cultural values vis-à-vis approaches to end FGC

The community provides multi-layered cultural reasons for practicing the FGC that are intended to challenge the government's intervention efforts to stop it. The main cultural explanations they provide include sexual modesty (controlling women's sexual drive), preserving premarital virginity, and developing accepted feminine behavior that ensures the marriageability of the girls. The community uses the compound term "*dhaqnaqabaa*" or "*huuba irraa fuudhuu*" to refer to both FGC and male circumcision. The term *dhaqna* refers to the body, and *qabaa*, from *qabuu*, means to 'catch'. Literally, it means 'catch the body' (to circumcise). According to our participants [36, 40–42] and FGD [37], a woman who has not undergone FGC is clean neither physically nor ritually, for her body is not caught for cleaning and cleansing.

This fact becomes clearer by the phrase "*huuba irraa fuudhuu*," which means to "remove something unwanted." The term "*huuba*" means 'unwanted', or 'smut'; *irraa fuudhuu*, means "to pick" or "remove something from a surface." Together, the two words mean "eliminate a chip or something malignant or unwanted." The phrase "*huuba irraa fuudhuu*" also implies that female genital cutting is as important as male circumcision [43]. A male person who is not circumcised is mocked at and insulted with a derogatory word 'uncircumcised' or 'one whose prepuce is still intact.' The expression shows that there is something alien and, therefore, unwanted on the genitalia, and due to this, should be removed. For the community, FGC is the surgical removal of the phallic and "unnecessary" or "unwanted" part of the genitalia, the clitoris, to make her a real woman, both sexually and physically, because the phallic is for men, and therefore, should be removed from the females.

Beyond the physical concern, the surgery entails a behavioral change that ensures femininity in terms of gender. There is an assumption that girls who have not undergone FGC are hypersexual, unclean, and indulging. As a result, our data from primary sources unanimously show that girls who have not undergone FGC are not esteemed and wanted for marriage. This is so because they still have the erectile part of their genitals (the clitoris), have not yet attained femaleness physically, and are not also properly feminine in their gender.

The community uses different sayings to ridicule women who have not yet undergone genital cutting. Referring to the physical presence of the clitoris, they say *ishiin eeboo baatti, dankaraa baattee deemti*, 'she carries a spear'. Focusing on the sexual behavior and gender identity of the uncircumcised girl, the community often uses such sayings as *ishiin farda*,'she becomes disorderly (like a horse);' *safuu dhabdi*, 'she is unethical;' *ija re'ee nyaatti*, 'she becomes shameless;' *abbaa manaa jala hin bultu*, 'she cannot remain with one husband'; *dhaabbattee hafti*, 'she will remain a bachelor' [37, 44]. Any act deviating from this norm has social costs, including ostracization, in which girls are labeled as being 'unclean', 'clumsy', 'unfit' for domestic work, or 'having uncontrolled sexuality', and thus unmarriageable [4].

Men do not choose such girls for marriage, mainly because of the fear of hyper-sexuality and untidiness. If they get married, their marital life will not be successful. One of our participants, a woman, tried to substantiate this view by narrating the story of a certain unsuccessful marriage. She knew an uncircumcised woman who married and divorced three times. According to this informant, the divorcee did not have a stable marital life, mainly because she did not undergo genital cutting and due to this she challenged those who married her [40]. We, the researchers, have not confirmed the story and do not know the real reasons behind the frequent divorce of the woman in focus. However, this informant, who is a pro-FGC strongly, recommends FGC as the best therapy to ensure a successful marriage for a woman and that is why she made all her daughters undergo the genital cutting.

Within the prevailing patriarchal cultural framework, women, as such, mothers and grandmothers, are the prime promoters of FGC because they are given the irreplaceable role of socializing and shaping, including the sexual behavior of girls, and one of the instruments for this is FGC. If a girl behaves in a customarily wrong way, the problem is soon attributed to the way she has been brought up. To express this fact, they use the common expression *isheen guddisa badde*, 'she was badly brought up.' If the misconduct is related to sexual activities, the blame becomes more severe and goes to the mother or the grandmother who brought her up.

A mother whose daughter has not undergone FGC often receives shaming remarks from others. Sayings such as *Intalli kee saafeela kan dhaabbattee hafte*, 'You are the mother of the uncircumcised, therefore, unmarried girl'. It is so shameful and dishonoring to be insulted in this way. In another saying, *haadha ilaali intala fuudhi*, 'see the mother before marrying her daughter', similar to the saying 'all daughters grow up to be like their mothers' [39, 44], members of the community convey the strong message to families that they have to cut the genitals of their daughters if they want them to have a successful marital life.

In substantiating this view, a father-informant whose wife organized a genital-cutting ceremony for their daughters without his consent, narrated the following story of his family. Our informant has two young daughters and claims that he is aware of the negative impacts of FGC because he completed grade ten, while his wife, who has no formal education, is a proponent of the practice and believes that it is mandatory and necessary for the cleanliness of women and cultural conformity. According to this informant, his wife believes that it is her responsibility to raise her daughters and mold them into ideal female behavior. If her daughters remain uncut, she would be responsible for the disgraceful state of the girls. Thus, she organized a secret genital-cutting session for her young daughters without the knowledge of the husband. She assigned her sister-in-law as a *jaala* (sponsor, to seat behind and cover the girls' eyes during the surgery) to keep the case confidential [45].

The father recognized lately that his daughters had undergone FGC. He noticed the girls receiving treatment while returning home from his daily farming activities. He strongly indicates that for a mother, letting her daughter remain with the clitoris is an incurable and lifelong psychological and moral failure. He says that *rakkoon dubartoota bira jira. Ani kanan dhaqna qabadhe maalan ta'e jedhu. Ani qofti mucaa koo dhaqna qabsiisuu dhiisee maalifan kan kolfaa ta'a; mucaa kootti kofalchiisa jedhu.* 'The problem is with the mothers. They say, though I had my genital cut, what has happened to me? Why do I make my daughter and myself mocked at by others leaving her uncut?' [45].

We were able to talk to the mother, who organized a secret session to cut her daughters' genitals. She proudly explained how she was successful and admitted that the report from her husband is true [46]. In relation to this, there is a nexus between parental education and values for FGC in Africa. Particularly, maternal education matters, and the more educated they are, the less they practice FGC on their daughters, and vice versa [26]. This is confirmed by the participating school teachers who have daughters but not performed FGC on them. Three female

teachers reported that they have not exposed their daughters to FGC, and promised not do so in the future [42, 47, 48]. The remaining two female teachers said that, although they are against the practice; they are not sure about what may happen when they will have daughters as the issue of FGC entails the interventions of several stakeholders [49, 50]. One of the male school teachers admitted that his elder daughter had already undergone FGC under the pressure of his mother, while the junior ones have not as he and his wife strongly resisted the urge [51]. The second male teacher, who has a young daughter of two years, said that he would not perform any FGC [52].

Coupled with this, the prevailing cultural norms put pressure on girls to have their genitals cut. Our data indicated that peer groups and friends harass, tease, and laugh at girls who have not undergone genital cutting [47, 48, 53, 54]. One of our participants, who claims that he is aware of the negative consequences of FGC, narrated to us how he finally submitted to the firm demand of his only daughter to have her genital cut. The father says that when she was eleven years old, she strongly demanded to undergo genital cutting; otherwise, she would stop going to school and escape somewhere. The father says that he was helpless to reject this demand from his daughter, for the denial might lead to a worst-case scenario, i.e., the disappearance of their daughter. In the meantime, he also believed that there was a hidden hand of his wife behind the pressure that their daughter was exerting. His wife did not openly oppose his position but agreed with her daughter and designed the way to achieve the final goal of cutting the genitalia [53].

## Withstanding the criminalization of FGC

Despite the campaign against the FGC by the government, the practice is still widespread, and the legal actions against it by the judiciary are close to nil. This is mainly because the community withstands the potential legal actions either by paying the imposed fine or by hiding the practice from legal or concerned bodies to escape being prosecuted [38, 44, 48, 52]. According to our participants, several factors contribute to this effect. Primarily, the law that forbids FGC lacks any deterrent effect. The law was enacted some years ago, and given the fast devaluation of the Ethiopian currency, a fine of five hundred Birr (as stated in Article 565 of the Ethiopian Criminal Code) has no preventive effect. One of our participants expressed the amount of the fine, saying, *dhibbi shan gatii indaanqoo tokkooti; eenyu illee adabbii qarshii dhibba shaniif jedhee dhaqnagabaa hindhiisu.* 'Five hundred Birr is just the price of a rooster; this penalty never deters anyone from practicing FGC' [45]. Regardless of its deterrent effect, our data indicate that the police, in cooperation with the Office of WCYA, brought a few cases of FGC before the district court and penalized the actors, including parents, sponsors, and circumcisers, as per the Ethiopian Criminal Code Article 565. Yet, parents always prefer to cover the fine instead of refraining from the practice because the payment is so meager that it can easily be afforded [38, 42, 44, 53, 55].

**Concealing FGC.** Alternatively, at best, parents hide FGC from legal personnel to escape prosecution. There are several ways to do this. Primarily, the local procedures have undergone a dramatic change in terms of the setting of the practice. The changes are in terms of the age at which girls get their genitals cut, the practitioners who do the surgery, the selection of the sponsors, the circumcision rituals, and the publicity of the practice. Data from interviews and FGDs indicate that there is no specific age at which girls get their genitals cut. It varies greatly based on parents' interests and readiness, convenient time to keep the practice confidential, external factors such as government intervention, and so on [36, 44, 56].

The community maintains the confidentiality of the practice by adjusting its timing, season, and occasion. Nowadays, parents subject their daughters to FGC at a young age, a few days or

months after delivery, when it is less likely for anyone other than immediate family members, the sponsor, and the practitioner to know about the procedure. This enables parents to maintain maximum confidentiality. Our informant from the office of WCYA [44] stated that cutting baby girls' genitalia at an early age is one of the challenges to stopping FGC. A pertinent study [9] shows that the age of genital cutting for more than 70% of Ethiopian girls is below 10 years, and about 49% have undergone genital cutting before they celebrate their 5th year. Parents pretend that they do not perform FGC; individuals in rural Ethiopia, particularly older ones, are in favor of FGC while concealing their true support for it [24].

Secondly, parents arrange *jaalaa* (sponsor) from the same family or close relatives to keep the practice private. Traditionally, sponsors come from outside the family because it is one way of establishing fictive kinship ties. Under the current scenario, priority is given to performing genital cutting in favor of the required cultural conformity rather than establishing kinship ties [57].

The third way of hiding FGC is by confusing it with other religious or non-religious festivities. This is common during the week after the Ethiopian Easter in April or early May every year. During this time, Ethiopian Orthodox Church followers, who are the majority in the area, end the two-month-long fasting period. The season is also the most joyful religious and cultural occasion when people visit and invite each other. The festivities and the celebrations then create a good opportunity for the people to camouflage the FGC with the occasion. Similarly, other non-religious ceremonies and occasions, like marriage, also serve to cover up the FGC [36, 37, 44].

The fourth way to keep FGC confidential is by confusing it with male circumcision. This is easy when the parents have boys and girls who are ready for circumcision. Parents make the necessary preparations officially to circumcise boys. Covertly, the same ceremony is used for the FGCs of their daughters. In some cases, parents use a male circumciser to do FGC. The use of male circumcisers for FGC is indeed a new phenomenon [42, 44].

When such confusion happens, the circumcised boys appear at the front bed, where visitors can see, talk, bless, and encourage them. Neighbors and relatives pay a visit to the circumcised to wish him a speedy recovery from the wound. They usually say, *"Madaan madaa saree siif haa ta'u,"* meaning, "Let your wound be like that of a dog" (naturally, a dog's wound is healed quickly because it is kept clean by licking). Girls stay in the backyards to be out of sight of the visitors. The family pretends that the entire ceremony is held for the circumcision of the boys. The fifth alternative way of hiding FGC is by taking girls to another place in the name of visiting their relatives. This is more common for urban dwellers who are not confident enough to cut the genitals of their girls for fear of being prosecuted or being blamed by their friends. In this case, the confidentiality of the cutting can be easily maintained from the outside, as the girls are guests there [44].

A study in Senegal [33] also reported that the practice of FGC was concealed to avoid being prosecuted. Under the threat of legal measures, people have two options: either to stop practicing it or to continue doing it secretly. It is less likely that the fear of prosecution can lead to the abandonment of FGC; rather, individuals may opt for other alternatives to decrease or escape the risk of being prosecuted.

Our finding also shows that FGC has the paradoxical effect of concealing and disclosing at the same time. Parents try to hide the act from the police and WCYA offices to escape legal prosecution. On the contrary, they also want the exercise to be known to the public at a certain point to secure cultural approval [44, 47, 48, 52, 55]. The parents want their friends and neighbors to witness the practice so that any potential teasing or mockery by others could be averted. With this, mothers also make themselves free from being blamed for letting their daughters go with an "unwanted" [56, 58, 59].

**Lack of collaboration.** The more solid challenge to stopping FGC is the lack of cooperation between members of the community and the concerned offices, notably the police and the Office of WCYA, the two offices that primarily work on the abolishment of the practice of FGC. However, they rarely access information on any planned and/or practiced FGC in the district. This is because the community considers FGC as a normal practice that does not demand the intervention of the police or any other government body. For the larger community, FGC cannot be any good reason to expose one's neighbors, friends, or relatives to any legal prosecution. One of our participants stated that *warri mucaa isaanii yoo dhaqna qabatan, miidhaas ta'ee faayidaan kanuma isaaniti. Isaan caalaa mucaa isaanii kan jaalatu eenyuu? Maaltu poolisii bira nama geessa*? 'If the parents do FGC for their daughter, the advantages and disadvantages will be theirs. Who else can care for a child more than the parents? What reason can take one to the police?' [45]. In the absence of any information from the community, therefore, it is less likely that the government body will intervene to tackle the FGC [38].

In addition, data from the police office indicate some challenges to enforcing the law to prohibit FGC. These include logistics problems, poor infrastructure, a lack of budget, and other facilities. For instance, a shortage of transportation facilities to reach every corner of the district prohibits the police from aborting a planned FGC or reacting to legal measures on time [38, 44, 56, 59]. There is also a paradox in the execution of the law against FGC. Usually, when the police identify a case of FGC, they take the injured girls to health centers for medical treatment and arrest the parents for committing the act. The questions are, "Who must take care of the victims if the parents are under police custody? Who should afford the required expenses for medication? Where the victims should be accommodated? What transportation facilities are available, and who should provide the services for the girls who have already been injured and cannot walk on foot?" Since the office of the police does not have the logistics and budget for these services, they usually become reluctant to imprison or impeach the parents.

A participant from the police office narrated one instance as follows: in May 2019, the police found four girls who underwent genital cutting. The parents were arrested, and the girls were taken to the health center for medical checkups and treatment of their wounds. The health center provides only outpatient services. The police staff individually contributed to cover the expenses for medication, as there is no government budget for this kind of service. Finally, the police had to spot where to keep the girls. Because they needed all kinds of support, like care for their wounds, what to eat and drink, and someone to look after them since their parents were already detained. Finally, the police decided to free the parents and allowed them to take care of their daughters [38].

## Discussion

The local phrase *dhaqnaqabaa* stands both for male and female circumcisions. A comparison between male circumcision and FGC is a tendency to normalize the latter [9]. Our findings reveal that the public perceives circumcision as a normal surgical procedure intending to remove an unnecessary thing both from male and female genitalia. The phrase *huuba irraa fuudhuu* in particular suggests that both male circumcision and FGC are important since they remove the unwanted part of the genitals through an easy procedure. Thus, the community has a common term and similar values for the circumcision of both sexes.

For the community, FGC is valued for physical cleanliness, sexual decency, gender construction, and marriageability. These findings partly corroborate pertinent work (e.g. [2]), which shows that some of the justifications behind FGC include the preservation of virginity, ensuring fertility, rite of passage, family honor, marriageability, hygiene and cleanliness, infant survival, increasing sexual pleasure for the male, and religious observance. Our findings,

however, did not show any explanation related to religion, infant survival, fertility, or male sexual pleasure for practicing FGC in the research area. As far as the nexus between religion and FGC is concerned, although the people predominantly ascribe to Orthodox Christianity, they do not associate the practice with this religion in any case, rather itis deeply embedded in their day-to-day lives.

This study shows that the overall cultural framework that favors FGC is patriarchy. This is more vivid in terms of what is labeled as the "clitoral economy [20]." This concept depicts the balance between the female body (clitoris) and that of the male's (penis). The study reveals that the culture bestows the clitoris with special symbolic power over the female body. The community refers to the clitoris as phallic, which challenges the hegemonic, phallic economy of men. Since the culture tends to preserve something phallic only for males, FGC is thought to be the remedy to subdue the clitoris that bears an erectile feature. This is typical patriarchy, which works to control women's sexuality to the best benefit of the males.

Generally, patriarchy works behind the scenes, and men are not at the forefront of promoting FGC. That means women are the prime promoters of the practice, for they are the products of the longstanding patriarchal values. Therefore, to understand the value of FGC, it is important to consider the role of women and their perceptions of ideal women's bodies, beautifications, and body modifications within the framework of the patriarchal values. Women want to shape the sexual and moral behavior of their daughters so that they will meet the ideal standard of womanhood. Our finding corroborates a study [21], which explicitly declares that the causal explanation behind the survival of FGC is not only patriarchy which provides the guiding principles, but also other products of the patriarchal culture itself.

The study shows that the participating mothers strongly supported FGC for their daughters, even when their husbands were reluctant about it. This is because what matters is not the position of individual husbands but rather the wider patriarchal cultural setting. The findings further indicate that the cultural framework in which they act—patriarchy—imposes mothers' absolute responsibilities in shaping the behavior of their daughters, including their sexual behavior. This finding confirms the position of some international organizations (e.g. [4]), which shows that the prevalence of FGC depends more on some background characteristics. Less educated girls and women in rural areas are more exposed to FGC. The finding is also relevant to some works [5, 26, 27], which show the nexus between maternal education and the FGC. This study shows that resistance comes from uneducated mothers, at least, while the husbands who claim to be aware of the negative consequences of FGC are ready to conform to the change. This fact is further substantiated by our field data, where educated mothers, such as school teachers, decided to leave the practice behind.

The government attempted to bring FGC to an end through both health and legislative approaches. The health and psychological effects of FGC emphasize women's inalienable rights, especially bodily integrity and sexual pleasure [31]. Our findings show that the local community is aware of the perceived negative health effects of FGC, mainly the risk of infection and childbirth complications. The sexual integrity approach has not been explicit in the local efforts to stop FGC since the custom does not allow the discussion of the topic as a public agenda to justify the end of FGC.

However, the community does not accept the health approach at face value. Particularly, older women strongly challenge and disqualify the scientific explanations and justifications given regarding this point. They often refer to their lived experiences to override the stated negative effects of the practice. This finding also corroborates the studies [22, 23] that discuss how older women act as active agents to reinforce patriarchal traditions by influencing younger girls to abide by the traditions they have been through.

Our finding also supports the view that the bodily and sexual integrity approach is challenged because sexual enjoyment is also cultural. Cultural values remained strong enough to withstand the theoretical orientation and awareness-creation of various agents [28]. The issue of sexual pleasure is not part of the discussion either to justify or resist FGC. Similarly, the human rights approach is neither fully understood nor is part of the discourse of the general public. The overall practices suggest the precedence of cultural values over private or individual rights.

The Ethiopian Criminal Code has labeled FGC as an illegal act. Some news of prosecution and the threat of trial are also public. Yet the commandment has borne no fruit; rather, empirical instances show that the practice of genital cutting has persisted. FGC is illegal only from a legal perspective; it does not win public acceptance and recognition in practice. The legislative approach has faced serious challenges and caused unintended adverse effects for the act, inviting less experienced female or male practitioners to do the cutting and for the parents to refrain from seeking medical treatment for the recipient's medical complications. This finding substantiates a study (e.g. [1]) that shows how parents practice FGC underground and abstain from seeking health care for any complications that may result from the surgery. Similarly, medical personnel also fear to treat the victims who received FGC and abstain from getting involved in it.

The effectiveness of the legislative approach in changing people's behaviors to end FGC is also challenging [28]. Female genital cutting is prevalent in many countries where laws prohibit it because the practice is conducted secretly [16]. Our finding supports these views in that the legislative approach lacks deterrent effects. Firstly, only a few individuals were prosecuted for cutting girls' genitalia. This seems common in other parts of the world, particularly in several African and Asian countries [2]. A study on FGC in the northern and northeastern parts of Ethiopia reported that many such cases remained unreported for the reason that the community performs it secretly in the distance and remote areas where the judiciary might not have access to information or secure evidence to prosecute the offenders and enforce the law [5]. Secondly, our finding indicates that the deterrent effect of these legal measures is under question, for the fine is so meager for everyone to afford. Mothers are ready to pay whatever the price could be rather than being ashamed by letting their daughters remain uncut. This could be why about 90% of schoolgirls who participated in the study at the same site reported that they had already undergone FGC [3]. Moreover, the very idea of banning a cultural practice through a legislative approach has its own challenge. This way of attempting to end a cultural practice causes a feeling of top-down imposition and faces community resistance [31].

## Conclusion and recommendations

To conclude, despite the several efforts by stakeholders to terminate FGC, it has survived the test of time as a longstanding tradition in the area. The multi-layered cultural explanations the people provide include sexual modesty, preservation of premarital virginity, achieving a feminine body feature and the ideal feminine behavior, and ensuring the marriageability of the girls. In short, FGC is a deeply-rooted socio-cultural practice that is related to the prevailing patriarchal system and its products, such as the sexual control of men over women, and gender roles and positions. On the other hand, several efforts have failed to bear fruit because they are framed in the global conceptual and institutional contexts that disregard the local cultural values and institutional setups. In addition, laws and rules that forbid this practice are not only weak but also face several financial, logistical, and practical challenges for implementation.

Thus, based on our findings, we recommend the following: Primarily, the stakeholders better engage local leaders (of both sexes) and use their knowledge in their efforts to terminate

FGC. New ideas and approaches, if integrated into the local knowledge, become more effective, cemented, and face less challenges and contradictions. Secondly, policymakers, program managers, educators, and other relevant bodies should work hand-in-hand in developing effective intervention mechanisms to terminate FGC. To this end, understanding and addressing those factors that strengthen the patriarchal values and its products among the community have to be effectively addressed. Thirdly, it is pertinent to enhance and utilize the law both in terms of promulgation and implementation. This can be achieved by increasing the fines, improving law enforcement through active case finding, and providing appropriate care and shelter for FGC victims once they are rescued. Fourthly, combating FGC requires allocating resources for advocacy, policy change and enforcement, awareness-raising, and engagement of civil society, parents, local elders, and religious leaders. Fifth, in Ethiopia, although there is no regional state where FGC came to an end, the ONRS can draw lessons from regions where the prevalence of this practice is lesser. Finally, the findings suggest that working closely with the community and educating girls about the multi-faceted implications of FGC are viable ways of ending it.

## Supporting information

**S1 File.**
(DOCX)

## Acknowledgments

We would like to express our special thanks and gratitude to the participants who provided us with valuable data to write this article. We also owe thanks to the local government officials in the district for cooperating with us to conduct this research.

## Author Contributions

**Conceptualization:** Zerihun Mekuria Tesfaye, Dejene Gemechu Chala.

**Data curation:** Zerihun Mekuria Tesfaye, Dejene Gemechu Chala.

**Formal analysis:** Zerihun Mekuria Tesfaye, Dejene Gemechu Chala.

**Investigation:** Zerihun Mekuria Tesfaye, Dejene Gemechu Chala.

**Methodology:** Zerihun Mekuria Tesfaye, Dejene Gemechu Chala.

**Project administration:** Dejene Gemechu Chala.

**Validation:** Dejene Gemechu Chala, Jira Mekonnen Choroke.

**Writing – original draft:** Zerihun Mekuria Tesfaye, Dejene Gemechu Chala.

**Writing – review & editing:** Dejene Gemechu Chala, Jira Mekonnen Choroke.

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
