## [Decision Letter · Decision Letter 0]

1 Aug 2024

PONE-D-24-24076Forbidden, yet common: female genital cutting among the Oromo in central EthiopiaPLOS ONE

Dear Dr. Chala,

Thank you for submitting your manuscript to PLOS ONE. After careful consideration, we feel that it has merit but does not fully meet PLOS ONE’s publication criteria as it currently stands. Therefore, we invite you to submit a revised version of the manuscript that addresses the points raised during the review process.

We look forward to receiving your revised manuscript.

Kind regards,

Joyce Jebet Cheptum

Academic Editor

PLOS ONE

Journal Requirements:

2. In the online submission form, you indicated that the data are available with the researchers and may be provided upon requests.

Additional Editor Comments:

The paper is well written, however, the description of the study design is not clear. There is a mention of secondary data being used, however, it is not clear how this data was retrieved, the source, how it was accessed etc.

There should be a clear description of the link between primary and secondary data.

Reviewers' comments:

Reviewer's Responses to Questions

**Comments to the Author**

1. Is the manuscript technically sound, and do the data support the conclusions?

Reviewer #1: Yes

Reviewer #2: Partly

2. Has the statistical analysis been performed appropriately and rigorously? 

Reviewer #1: Yes

Reviewer #2: No

3. Have the authors made all data underlying the findings in their manuscript fully available?

Reviewer #1: Yes

Reviewer #2: No

4. Is the manuscript presented in an intelligible fashion and written in standard English?

Reviewer #1: Yes

Reviewer #2: Yes

5. Review Comments to the Author

Reviewer #1: The manuscript is very well written. It highlights an age-old traditional harmful practice - FGC, confined to a few communities, depicting the sad situation and the system's inability and weakness in addressing it.

It's unfortunate that the developed world and other regions have not been able to support these girls and the countries where FGC is still prevalent in eradicating this harmful practice. FGC not only affects the physical health of the girls but also their mental and psychological well-being.

Based on qualitative data. The authors have done an excellent job designing, organising, and interpreting the qualitative study to identify all significant risks and associated factors for FGC. Please take note of the following suggestions and comments:

1. Consider adding a paragraph at the end of the manuscript that provides recommendations or conclusions based on the study's findings. This can guide policymakers, program managers, and donors in developing effective interventions to eliminate FGC.

2. While the authors effectively concluded that patriarchy and male dominance underpin FGC, it's essential to acknowledge that patriarchy is deeply entrenched in societies. Understanding and addressing the factors that strengthen or weaken the patriarchal system can influence the practice of FGC.

3. The paper could benefit from discussing the role of religion and religious leaders in FGC.

4. Consider comparing the findings with quantitative data from the available studies.

5. One of the key recommendations from your study is to strengthen the law regarding Female Genital Cutting (FGC). This can be achieved by making the law more powerful and effective as a deterrent to those who promote or practice FGC. It can be achieved by increasing fines or punishments, improving law enforcement through active case finding, and providing appropriate care and shelter to FGC victims once they are rescued.

6. Combating FGC requires allocating resources for advocacy, policy change, enforcement, awareness-raising, and engagement of civil society, parents, girls opposing FGC, and religious leaders.

7. Giving examples from regions where FGC has been eradicated and highlighting successful strategies would be beneficial.

Reviewer #2: I wanted to commend the authors for having a such amazing article. Having said that, I recommend to revise this publishable paper considering a revision on its write up using the IMRAD style. I found components of the method section in the introduction section. I would like also to forward my question regarding the data analysis method you have applied. What was your study objective? Which study design did you apply? And Which qualitative data analysis method was utilized? Thank you!

6. PLOS authors have the option to publish the peer review history of their article (what does this mean?). If published, this will include your full peer review and any attached files.

Reviewer #1: **Yes: **Dr Amrita Kansal, M.B.B.S., M.D. (Community Medicine); Public Health Expert; Technical Officer WHO - SEARO

Reviewer #2: No

---

## [Author Response · Author response to Decision Letter 0]

10 Oct 2024

We have thoroughly addressed the reviewers' comments and uploaded them as file.

---

## [Decision Letter · Decision Letter 1]

28 Nov 2024

Forbidden, yet common: female genital cutting among the Oromo in central Ethiopia

PONE-D-24-24076R1

Dear Dr. Chala,

We’re pleased to inform you that your manuscript has been judged scientifically suitable for publication and will be formally accepted for publication once it meets all outstanding technical requirements.

Kind regards,

Joyce Jebet Cheptum

Academic Editor

PLOS ONE

Additional Editor Comments (optional):

Reviewers' comments:

Reviewer's Responses to Questions

**Comments to the Author**

1. If the authors have adequately addressed your comments raised in a previous round of review and you feel that this manuscript is now acceptable for publication, you may indicate that here to bypass the “Comments to the Author” section, enter your conflict of interest statement in the “Confidential to Editor” section, and submit your "Accept" recommendation.

Reviewer #1: All comments have been addressed

Reviewer #2: All comments have been addressed

2. Is the manuscript technically sound, and do the data support the conclusions?

Reviewer #1: Yes

Reviewer #2: Yes

3. Has the statistical analysis been performed appropriately and rigorously? 

Reviewer #1: Yes

Reviewer #2: N/A

4. Have the authors made all data underlying the findings in their manuscript fully available?

Reviewer #1: Yes

Reviewer #2: Yes

5. Is the manuscript presented in an intelligible fashion and written in standard English?

Reviewer #1: Yes

Reviewer #2: Yes

6. Review Comments to the Author

Reviewer #1: The paper on female genital cutting (FGC) is well-written and presents valuable data along with relevant sociocultural factors that explain why this forbidden and highly unacceptable practice continues to be followed by communities in various parts of the world. The quantitative analysis demonstrates how the interplay between different beliefs and practices contributes to the continued prevalence of FGC in certain communities. This information can guide policymakers in strengthening the enforcement of laws aimed at eliminating FGC and protecting women's rights.

Reviewer #2: (No Response)

7. PLOS authors have the option to publish the peer review history of their article (what does this mean?). If published, this will include your full peer review and any attached files.

Reviewer #1: **Yes: **Amrita Kansal

Reviewer #2: No

---

## [Editor Report · Acceptance letter]

3 Dec 2024

PONE-D-24-24076R1 

PLOS ONE

Dear Dr. Chala, 

I'm pleased to inform you that your manuscript has been deemed suitable for publication in PLOS ONE. Congratulations! Your manuscript is now being handed over to our production team.

Kind regards, 

on behalf of

Dr. Joyce Jebet Cheptum 

Academic Editor

PLOS ONE